# A Novel Hybrid Nanosystem Integrating Cytotoxic and Magnetic Properties as a Tool to Potentiate Melanoma Therapy

**DOI:** 10.3390/nano10040693

**Published:** 2020-04-06

**Authors:** Nuno Cruz, Jacinta Oliveira Pinho, Graça Soveral, Lia Ascensão, Nuno Matela, Catarina Reis, Maria Manuela Gaspar

**Affiliations:** 1Instituto de Biofísica e Engenharia Biomédica (IBEB), Faculdade de Ciências, Universidade de Lisboa, Campo Grande, 1749-016 Lisboa, Portugal; nunocruz9009@gmail.com (N.C.); nmatela@fc.ul.pt (N.M.); 2Research Institute for Medicines, iMed.ULisboa, Faculty of Pharmacy, Universidade de Lisboa, Av. Prof. Gama Pinto, 1649-003 Lisboa, Portugal; jopinho@ff.ulisboa.pt (J.O.P.); gsoveral@ff.ulisboa.pt (G.S.); 3Centro de Estudos do Ambiente e do Mar (CESAM), Faculdade de Ciências, Universidade de Lisboa, Campo Grande, 1749-016 Lisboa, Portugal; lmpsousa@fc.ul.pt

**Keywords:** Cuphen, iron oxide nanoparticles, pH-sensitive liposomes, magnetic targeting, melanoma therapy, nanoparticles safety

## Abstract

Cancer is a major health concern and the prognosis is often poor. Significant advances in nanotechnology are now driving a revolution in cancer detection and treatment. The goal of this study was to develop a novel hybrid nanosystem for melanoma treatment, integrating therapeutic and magnetic targeting modalities. Hence, we designed long circulating and pH-sensitive liposomes loading both dichloro(1,10-phenanthroline) copper (II) (Cuphen), a cytotoxic metallodrug, and iron oxide nanoparticles (IONPs). The synthetized IONPs were characterized by transmission electron microscopy and dynamic light scattering. Lipid-based nanoformulations were prepared by the dehydration rehydration method, followed by an extrusion step for reducing and homogenizing the mean size. Liposomes were characterized in terms of incorporation parameters and mean size. High Cuphen loadings were obtained and the presence of IONPs slightly reduced Cuphen incorporation parameters. Cuphen antiproliferative properties were preserved after association to liposomes and IONPs (at 2 mg/mL) did not interfere on cellular proliferation of murine and human melanoma cell lines. Moreover, the developed nanoformulations displayed magnetic properties. The absence of hemolytic activity for formulations under study demonstrated their safety for parenteral administration. In conclusion, a lipid-based nanosystem loading the cytotoxic metallodrug, Cuphen, and displaying magnetic properties was successfully designed.

## 1. Introduction

Melanoma, which derives from the malignant transformation of melanocytes, is one of most aggressive forms of skin cancer [1]. Treatment options are primarily based on surgery, radiotherapy and chemotherapy. When the disease is detected at an early stage, the cure is often possible with surgery. However, when it progresses to the metastatic phase, the prognosis is poor, with low survival rates, and therapy usually fails. In these particular situations, combined systemic approaches are implemented, including targeted or non-targeted immunotherapy, chemo or and/or hormonal therapies [2,3,4]. Despite some improved clinical responses, the survival rates are still very low [5,6], motivating the development of novel or more effective therapeutic approaches.

The clinical success of cisplatin, a Pt(II) complex, against several tumors has prompted the investment on coordination chemistry of metal-based drugs for cancer [7]. In recent years, metal-based compounds, namely gold and copper complexes [8,9,10,11,12,13,14,15], have been researched as anticancer agents. Copper–phenanthroline compounds were shown to promote oxidative DNA damage [16], and Cu^2+^ compounds, including dichloro(1,10-phenanthroline) copper (II), Cuphen have also been reported as inhibitors of aquaporin-3 (AQP3), an aquaglyceroporin that is overexpressed in several carcinomas including melanoma [17,18]. Previously, Cuphen has demonstrated potent in vitro cytotoxic effects against murine and human melanoma cells lines, with an half-inhibitory concentration under 3.5 µM in all tested lines [8,9]. Considering the in vivo setting, to minimize potential unwanted side effects in non-affected and to protect the metallodrug from premature degradation, Cuphen was associated with long circulating liposomes, a well-known drug delivery nanosystem [8,9]. Studies in healthy mice revealed no toxic effects after parenteral administration of Cuphen liposomes, demonstrating their safety [8].

The use of nanosized liposomes with prolonged circulation times in the blood is advantageous for targeting tumor sites, as the possibility to extravasate to affected areas increases and due to a deficient lymphatic drainage, liposomes accumulate. This phenomenon has been known as enhanced permeation and retention (EPR) effect [19,20,21]. In addition, solid tumors, such as melanoma, are characterized by a slightly acidic microenvironment, displaying pH values around 6 [22,23]. In order to take advantage of this feature and promote a local pH-triggered Cuphen release, Pinho and colleagues [9] evaluated the therapeutic effect of Cuphen liposomal formulations in a murine melanoma model, comparing pH-sensitive with non pH sensitive nanoformulations. The results showed a significant inhibition of tumor growth progression, compared to control group and animals receiving Cuphen in free form [9].

Following these promising results, the next step was to further enhance the accumulation and of Cuphen liposomes at melanoma sites through magnetic targeting. Magnetic iron oxide nanoparticles (IONPs) have been used for several applications in medicine, including cancer treatment via thermal ablation or hyperthermia [24], theranostics [25], as well as magnetically targeted drug carriers [26,27,28]. Hence, in order to potentiate melanoma therapy, the aim of the present study was to develop a new hybrid lipid-based nanosystem co-loading Cuphen, the cytotoxic metallodrug, and IONPs, for magnetic targeting as schematically demonstrated in Figure 1.

## 2. Materials and Methods

### 2.1. Chemicals

Hydrazine Monohydrate (H_4_N_2_.H_2_O), Dextran-70 (H(C_6_H_10_O_5_)_70_OH) and Ferric Chloride (FeCl_3_) were obtained from Sigma (Sigma-Aldrich, St. Louis, MO, USA). The lipids dimiristoyl phosphatidyl choline (DMPC), cholesteryl hemisuccinate (CHEMS) and distearoyl phosphatidylethanolamine covalently linked to poly (ethylene glycol) 2000 (DSPE-PEG) were obtained from Avanti Polar Lipids (AL, USA). Dichloro(1,10-phenanthroline) copper (II) (Cuphen) was obtained from Sigma (Sigma-Aldrich, St. Louis, MO, USA). All other reagents were of analytical grade.

### 2.2. Cell Line Culture Conditions

B16F10 and MNT-1 cell lines were maintained in Dulbecco’s Modified Eagle’s medium (DMEM) with high glucose (4500 mg/L), supplemented with 10% fetal bovine serum and 100 IU/mL of penicillin and 100 μg/mL streptomycin. 

### 2.3. IONPs Synthesis

Dextran-coated IONPs were produced following a previously described method [29], with some modifications in order to allow the use of Dextran-70 instead of Dextran-10 [30]. To produce Dextran-70 coated IONPs, FeCl_3_ (76 mg) and Dextran-70 (100 mg) were added to 8 mL of bidistilled water. The mixture was then placed into a reaction vial and stirred to promote dissolution. To the obtained solution, 1 mL of H_4_N_2_.H_2_O was added under stirring and, subsequently, placed into the Anton Paar Monowave 300 reaction chamber (Graz, Austria), at a temperature of 100 °C for 10 min. Uncoated particles were produced following the same procedure, except no Dextran-70 was added to the initial solution. The uncoated particles were separated from the solvent by centrifugation at 2200 *g* for 8 min (Sigma 2020-MK, St. Louis, MO, USA). The coated particles were subjected to dialysis using a dialysis sleeve (Medicell Int. LTD, London, UK, 12,000–14,000 MWCO) to remove any unreacted components remaining in suspension. The obtained IONPs were lyophilized during 24 h at −50 °C (freeze–dryer model, Edwards, CO, USA). Particle morphology and size were assessed by transmission electron microscopy (TEM) and dynamic laser scattering (DLS), respectively.

### 2.4. IONPs Characterization by TEM and DLS

Samples of Dextran-70 coated and uncoated IONPs were prepared for morphological analysis through the negative staining method. IONPs were resuspended in distilled water and droplets (10 μL) were placed on Formvar-carbon-coated grids. After a few minutes, during which the particles attach to the Formvar-carbon film, the grids were partially dried with a piece of filter paper. The material was then negatively stained with 1% uranyl acetate and left to dry at room temperature. Observations were carried out on a JEOL 1200EX transmission electron microscope (JEOL Ltd., Tokyo, Japan) at an accelerating voltage of 80 kV. Images were recorded digitally. The mean hydrodynamic size and polydispersivity index (PDI) of the IONPs was assessed by the DLS equipment Zetasizer Nano S (Malvern Instruments, Inc., Malvern, UK).

### 2.5. Preparation of Liposomes

Long circulating and pH-sensitive liposomes with the lipid composition composed of dimiristoyl phosphatidyl choline (DMPC), cholesteryl hemisuccinate (CHEMS), and distearoyl phosphatidylethanolamine covalently linked to poly (ethylene glycol) 2000 (DSPE-PEG), DMPC:CHEMS:DSPE-PEG, at a molar ratio of 57:38:5, were prepared by the dehydration-rehydration method [8,9,31]. An initial lipid concentration of 30 μmol/mL was used. Briefly, the selected phospholipids were dissolved in chloroform in a round-bottomed flask. The obtained lipid solution was evaporated (Buchi R-200 rotary evaporator, Flawil, Switzerland) to form a thin lipid film, which was then dispersed with (i) a Cuphen aqueous solution (750 μM) and (ii) a Cuphen and Dextran-70 coated IONPs aqueous solution at a final concentration of 750 μM and 2 mg/mL, respectively. The so-formed suspensions were frozen (−70 °C) and lyophilized overnight. The lyophilized products were rehydrated in HEPES buffer, pH 7.4 (10 mM HEPES, 145 mM NaCl) in two steps, to enhance compound incorporation [32]. Afterwards, all liposomal suspensions were filtered under nitrogen pressure (10–500 lb/in2) through polycarbonate membranes of proper pore size until the desired vesicle size was obtained, using an extruder apparatus (Lipex: Biomembranes Inc., Vancouver, BC, Canada). Non-incorporated Cuphen and IONPs were separated by gel filtration (BioRad Econo-Pac^®^ 10DG). The suspension was concentrated using a benchtop centrifuge at 15,000 *g* for 30 min (Sigma 2020-MK). In the case of Cuphen liposomes, the suspension was ultracentrifuged in a Beckman LM-80 ultracentrifuge (Beckman Instruments, Inc., Fullerton, CA, USA) at 250,000 *g*, for 120 min. The pellets were suspended in HEPES buffer. 

### 2.6. Liposomes Characterization

Liposomes were characterized in terms of Cuphen incorporation parameters and mean size by DLS (Zetasizer Nano S, Malvern, UK). Initial and final loading capacity was defined as the initial Cuphen to lipid ratio (Cuphen/Lip)_i_ and final Cuphen to lipid ratio (Cuphen/Lip)f, respectively. 

Cuphen was quantified spectrophotometrically at 270 nm (ε = 33,000 M-1 cm^–1^) after disruption of the liposomes with absolute ethanol [8,9]. The linearity of calibration curves was confirmed from 2.5 to 25 μM (R^2^ = 0.9998; Slope: 0.0324 ± 0.0001; y-intercept [x = 0]: −0.0036 ± 0.0006). Lipid content was determined using the method described by Rouser [33]. 

The determination of the percentage of the hybrid nanosystem that precipitates using a benchtop centrifuge or an ultracentrifuge was performed according to Equation (1):(1)$$\mathrm{Precipitated}\;\mathrm{Hybrid}\;\mathrm{nanosystem}\;(\%)=\frac{\mathrm{Lipid}\;\mathrm{in}\;\mathrm{pellet}\;(µ\mathrm{mol})}{\mathrm{Total}\;\mathrm{Lipid}\;(µ\mathrm{mol})}\times 100$$

### 2.7. Cytotoxicity Studies

Murine (B16F10) and human (MNT-1) cell viability was evaluated in the absence (control) or presence of increasing concentrations of Cuphen and Dextran-70 coated IONPs, alone or in combination, in free forms, by the MTT assay. Cells were placed in 96-well plates (200 μL), at a concentration of 5 × 10^4^ cells/mL and allowed to grow for 24 h, at 37 °C, 5% CO_2_ [9]. Next, complete culture medium was replaced, and cells treated with the formulations under study: (i) Cuphen at concentrations ranging from 0.5 to 7 µM; (ii) IONPs at concentration values from 1 to 7.5 mg/mL; and (iii) combination of Cuphen (1 and 5 µM) with IONPs at 2 mg/mL. Cells with only complete culture medium constituted the controls. Following the 24 h incubation period, complete medium was discarded and, to each well, 50 μL of MTT reagent (0.5 mg/mL in incomplete culture medium) was added. After an incubation period of 3–4 h, 100 μL of DMSO were added to each well to dissolve the formazan crystals. Absorbance was measured at 570 nm in a microplate reader (Model 680, Bio-Rad, Hercules, CA, USA). Cell proliferation was analyzed in GraphPad Prism^®^ 5 (GraphPad Software, San Diego, CA, USA). Values were plotted and fit to a standard inhibition log dose response curve to generate the IC_50_ values. A total of three independent experiments, with six replicates *per* condition, were carried out.

### 2.8. Magnetism Assays

For this in vitro test, NdFeB magnets (José Teixeira da Rocha, Unipessoal, Lda.; N38, stacked: 40 × 10 × 20 mm, 560.9 mT) were used. Briefly, 1.5 mL of Cuphen and IONPs liposomal suspension were placed in a 6-well plate. The magnet was positioned below the well containing the sample, at one of the extremities. Following the designated times of 1, 2, 4 and 19 h, magnetic exposure was ceased and samples (100 µL) from the magnet region and opposite extremity were collected. Cuphen contents were determined as described above. 

### 2.9. Hemolysis Assays

The hemolytic activity of Cuphen and Dextran-70 coated IONPs, in free and liposomal forms, was assessed using EDTA-preserved peripheral human blood [31], obtained from a voluntary donor. Serum was removed by centrifugation at 1000 *g* for 10 min and the erythrocyte suspension was washed three-times with PBS at 1000 *g* for 10 min. The formulations under study were diluted in PBS and distributed in 96-well plates (100 µL/well), for final concentrations ranging from 3.125 to 200 µM and 0.313 to 5 mg/mL, for Cuphen and IONPs, respectively. Then, 100 µL of erythrocyte suspension was added to all samples, followed by an incubation at 37 °C for 1 h and centrifugation at 800 *g* for 10 min. Supernatants were collected and the absorbance was measured at 550 nm, with a reference filter at 620 nm. The hemolytic activity in percentage for each tested sample was calculated comparing each individual determination to positive control (100% hemolysis, erythrocytes in distilled water) and negative control (erythrocytes in PBS), according to Equation (2): (2)$$\mathrm{Hemolysis}\;\left(\%\right)=\frac{\left(\mathrm{AbsS}-\mathrm{AbsN}\right)}{\left(\mathrm{AbsP}-\mathrm{AbsN}\right)}\;\times \;100,$$
where AbsS is the average absorbance of the sample, AbsN is the average absorbance of the negative control and AbsP is the average absorbance of the positive control.

## 3. Results

### 3.1. Characterization of IONPs

Dextran-coated and uncoated IONPs were produced following a previously described method [29], with some modifications. The obtained particles were characterized in terms of morphology by TEM and size by DLS for coated IONPs. As shown in Figure 2a, uncoated IONPs displayed an increased aggregation and a larger iron core compared to the Dextran-70 coated IONPs in Figure 2b, which were homogeneously dispersed. The same effect was observed by Oliveira, et al. 2019, when preparing uncoated IONPs [34]. Furthermore, since Dextran-70 was attached to the IONPs’ surface, the size of the whole particle was assessed by DLS, with values of 105 ± 2 nm and a PDI of 0.238 ± 0.009. In the following sections, Dextran-70 coated IONPs, the selected nanoparticles for the conducted studies, will be simply designated by IONPs.

### 3.2. Cytotoxicity of Cuphen and IONPs

The antiproliferative activity of free Cuphen and IONPs, alone and in combination, was evaluated by MTT assay in human (MNT-1) and murine (B16F10) melanoma cell lines (Figure 3). Data in Figure 3b,c show that Cuphen decreased cell viability of melanoma cell lines in a dose-dependent manner. For the tested cell lines, IC_50_ values below the micromolar range (<10 µM) were obtained: 4.3 ± 0.1 µM and 5.7 ± 0.8 µM for MNT-1 and B16F10, respectively. Although a slight loss of cell viability for B16F10 was obtained for higher concentrations, Figure 3d, no appreciable cytotoxic effects were observed for IONPs concentrations of 1 and 2 mg/mL for both cell lines. According to these data, cell viability was evaluated after incubation with IONPs, at the selected concentration of 2 mg/mL, in combination with Cuphen at concentrations of 1 and 5 µM, Figure 3e. The results revealed that the presence of IONPs, at the tested concentration, did not affect the cytotoxic properties of Cuphen, thus rendering their co-loading in liposomes feasible.

### 3.3. Liposomes Co-Loading Cuphen and IONPs

Liposomes with the lipid composition DMPC:CHEMS:DSPE-PEG (57:38:5), which display pH-sensitive properties, were prepared by the dehydration-rehydration method [9]. On one hand, nanosized liposomes require long ultracentrifugation cycles (≥2 h, 250,000 *g*) to precipitate [8]. On the other hand, IONPs easily precipitate in short centrifugation cycles using a benchtop centrifuge. In this sense, the encapsulation of IONPs in liposomes was indirectly confirmed by quantification of lipid content (%) of liposomes that precipitated under the experimental conditions depicted in Table 1. A total amount of 10 µmol of lipid was used in all experiments and the content in the pellet was determined for both centrifugation conditions and expressed in µmol and percentage.

As shown in Table 1, more than 50% of liposomes precipitated by both short centrifugation cycles, confirming that IONPs were associated with the nanosystem. 

As both centrifugation conditions led to equivalent results, for nanoformulations co-loading Cuphen and IONPs, the centrifugation cycle of 15,000 *g* for 30 min was selected to evaluate if Cuphen liposomes with the same lipid composition would show the same pattern. As depicted in Figure 4a no pellet was obtained, reinforcing the fact that Cuphen liposomes would only precipitate when co-loaded with IONPs (Figure 4b).

### 3.4. Influence of IONPs on the Physicochemical Parameters of Cuphen Liposomes

Cuphen nanoformulations were prepared with and without IONPs and their presence on the physicochemical properties of Cuphen liposomes was evaluated. The obtained results are shown on Table 2. Aiming to maximize the loading of IONPs, the influence of two different liposomal formulations LIP A and LIP B, with sizes <200 nm and <300 nm, respectively were tested. In all the developed nanoformulations, the presence of IONPs affected Cuphen incorporation parameters, leading to a reduction on the loading capacity and an increase of the mean size. With IONPs, LIP A displayed a loading capacity of 22 nmol/µmol, contrasting with the values of 26 nmol/µmol for Cuphen liposomes. The same tendency was obtained for LIP B, with loading capacity values of 25 and 35 nmol/µmol, respectively. Although LIP A and LIP B were produced under the same experimental conditions, a systematic increase on the mean size of nanoformulations co-loading Cuphen and IONPs was observed. For LIP A and LIP B, an increase from 127 to 162 nm and from 236 to 277 nm was attained, respectively. Even though the quantification of iron content in developed formulations was not determined, the obtained results clearly demonstrate that IONPs are indeed associated with liposomes. In addition, for all studied nanoformulations, the PDI was below 0.2 confirming the presence of monodispersed formulations. According to literature, these results further certify that IONPs are associated with the nanosystem and not in the free form [35]. Finally, and as previously observed (Table 1; Figure 4b), the presence of IONPs led to a higher percentage of liposomes after a short centrifugation cycle (15,000 *g*, 30 min) as demonstrated in Table 2 by the high % of lipid: 59 vs. 47 and 80 vs. 69 for LIP A and LIP B, respectively.

### 3.5. Validation of Magnetic Properties of the Developed Nanoformulations

To ensure that Cuphen-loaded liposomes containing IONPs exhibited magnetic properties, a simple in vitro assay was carried out. Figure 5 shows the scheme of the prototype static model used. As described in the methods section, liposomes loading both Cuphen and IONPs were subjected to a magnetic field of 560.9 mT for 1, 2, 6, and 19 h in separated assays. For each time point, samples were collected in both extremities of the well, and the Cuphen content was determined by spectrophotometry. The obtained data are depicted in Figure 6. 

Results demonstrated that Cuphen liposomes with IONPs displayed magnetic properties following an exposure to a constant magnetic field, for up to 19 h. This was a time-dependent effect, as an increase on Cuphen concentration (in %) at the magnet site was observed over time. From 1 to 19 h, a 10-fold increase on Cuphen accumulation was obtained.

### 3.6. Hemocompatibility of Cuphen and IONPs

Following the successful development and characterization of liposomes co-loading Cuphen and IONPs, the evaluation of the in vivo safety was performed by an hemolysis assay using human red blood cells (hRBCs) [8,31]. The tested formulations were free IONPs and liposomes co-loading Cuphen and IONPs. The obtained results are summarized in Table 3. 

The obtained data demonstrated that both formulations had negligible hemolytic effects (<5%) at the tested concentrations, ranging from 0.313 to 5 mg/mL and 3.125 to 200 µM for IONPs and Cuphen liposomes containing IONPs, respectively.

## 4. Discussion

Nanotechnology offers innovative tools to solve fundamental problems in cancer management, namely the efficient delivery of drugs to the diseased areas to maximize the therapeutic effect. Lipid-based systems, namely liposomes, are one of the most studied and successful delivery systems, with several examples already approved for use in the clinic, undergoing clinical trials or under preclinical development [36,37,38]. The present study focused on melanoma treatment through a novel hybrid lipid-based nanosystem. To this end, liposomes with magnetic properties were prepared by passively encapsulating IONPs within the aqueous interior compartment, a method that has been employed by other research groups [39,40,41,42,43].

The process to produce IONPs was modified to use Dextran-70 instead of Dextran-10, since according to Paul and co-workers [30], the use of Dextran-70 coated IONPs would result in IONPs with size (21 nm) and magnetic properties comparable to those using reduced Dextran-10. The main advantage of this modified method would be to skip the 12 h step for Dextran reduction, a highly time-consuming reaction. If successful, the applied modifications in the protocol would allow for a faster and simpler IONPs production via a microwave-assisted reaction. The microscopic analysis of coated and uncoated IONPs clearly demonstrated the benefits of the coating, as uncoated IONPs presented more aggregation and a larger iron core, compared to the coated ones. Aggregation was expected, since the IONPs high surface area to volume ratio renders them especially vulnerable to Van der Waals forces [44]. Furthermore, other factors associated with the suspension medium, including charge shielding effects in saline buffers or protein adsorption in biological medium, may contribute to aggregation [45]. On the other hand, Dextran-70 coated IONPs displayed a more disperse morphology and well-defined IONPs. The obtained images are according to the research published by Osborne and colleagues [29], where TEM imaging of Dextran coated IONPs should show only the iron cores, since Dextran would be transparent in TEM [29]. DLS assays were also performed and the obtained hydrodynamic size of coated IONPs corroborated data obtained from microscopic analysis. 

The main goal of this study was to develop liposomes with cytotoxic (Cuphen) and magnetic (IONPs) properties. Considering their physical and chemical properties, IONPs can be used either as part of the tumor treatment (e.g.,: hyperthermia and photodynamic therapy), or for the magnetic targeting of nanosystems carrying the therapeutic drug [27,46,47]. In clinical practice, the potential cytotoxicity evaluation of IONPs is a major priority in order to avoid unpredictable interactions that could lead to unwanted side effects [47]. Few in vitro studies have been published and are often inconsistent, in vivo assays are scarce and human studies are almost inexistent. A balance between benefit and risk has to be carefully considered for each specific application [28,48,49,50,51]. Nevertheless, in the present research, two main aspects were taken into consideration when developing this hybrid nanosystem: (i) IONPs should not, by themselves, affect cell viability and (ii) IONPs presence should not influence Cuphen cytotoxicity towards tumoral cells. Using murine and human melanoma cell lines, cell viability studies showed that Cuphen cytotoxicity was preserved in the presence of IONPs. Moreover, IONPs at 2 mg/mL proved to be innocuous, in terms of cellular viability, and so this was hence the selected concentration to be used for the preparation of the hybrid nanosystem co-loading Cuphen and IONPs.

The magnetic properties of Cuphen liposomes loading IONPs were confirmed. The proof of concept was performed by exposing the hybrid nanosystem at constant magnetic field of 560.9 mT for each analysed time. This assay allowed to confirm that an accumulation of Cuphen at the magnet site was achieved, being time dependent.

The absence of hemolytic activity of IONPs, after incubation with human red blood cells (hRBCs), revealed their suitability for parenteral administration. In vivo studies in a syngeneic melanoma model comparing the antitumor effect of Cuphen liposomes and the hybrid nanosystem co-incorporating IONPs should be envisioned in the near future. 

## 5. Conclusions

Considering these preliminary data, the synthetized IONPs are safe for parenteral administration. Moreover, they did not interfere with cytotoxic activity of the metallodrug and when co-associated with Cuphen liposomes demonstrated magnetic properties. Further research involving a syngeneic murine melanoma model should be carried out to validate the effectiveness of this novel hybrid nanosystem and the potential therapeutic improvement of Cuphen liposomes.

## Figures and Tables

**Figure 1 nanomaterials-10-00693-f001:**
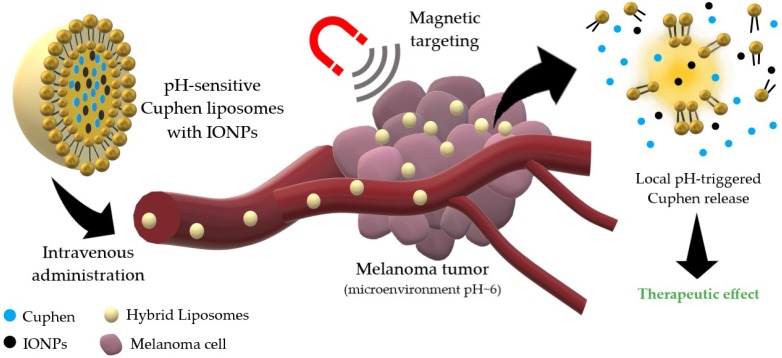
Schematic representation of the hybrid nanosystem integrating cytotoxic and magnetic properties as a tool to potentiate melanoma therapy.

**Figure 2 nanomaterials-10-00693-f002:**
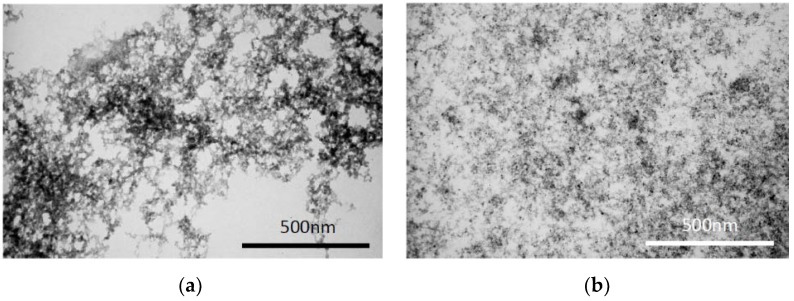
Representative TEM images of (**a**) uncoated iron oxide nanoparticles (IONPs) and (**b**) Dextran-70 coated IONPs. Scale bar = 500 nm.

**Figure 3 nanomaterials-10-00693-f003:**
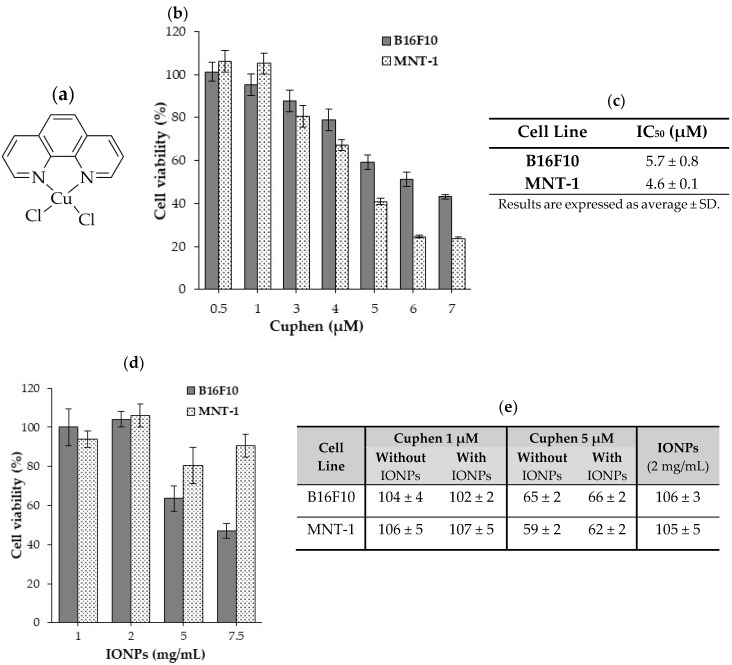
Evaluation of Cuphen and IONPs cytotoxic activity towards human (MNT-1) and murine (B16F10) melanoma cell lines. (**a**) Cuphen chemical structure; (**b**) cell viability after 24 h incubation with different Cuphen concentrations (0.5 to 7 µM); (**c**) Cuphen half-inhibitory concentration (IC_50_) values; (**d**) cell viability after 24 h incubation with IONPs at a concentration ranging from 1 to 7.5 mg/mL; (**e**) cell viability after a 24 h incubation period with Cuphen (1 and 5 µM) in the presence or absence of IONPs at a concentration of 2 mg/mL. Data are expressed as mean percentage (%) of control ± SD of three independent experiments with six replicates each.

**Figure 4 nanomaterials-10-00693-f004:**
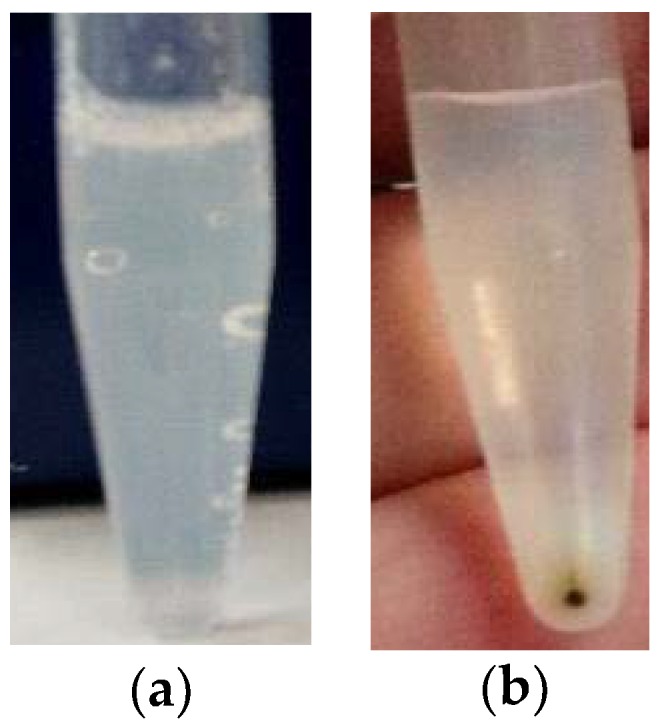
Representative images of (**a**) Cuphen liposomes and (**b**) liposomes co-loading Cuphen and IONPs, after a centrifugation cycle of 15,000 *g*, 30 min.

**Figure 5 nanomaterials-10-00693-f005:**
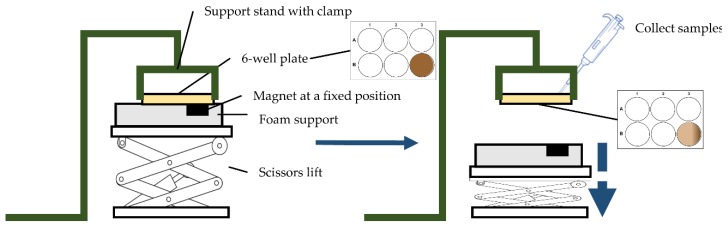
Scheme of the prototype static model used for validation of the magnetic properties of the developed nanoformulations.

**Figure 6 nanomaterials-10-00693-f006:**
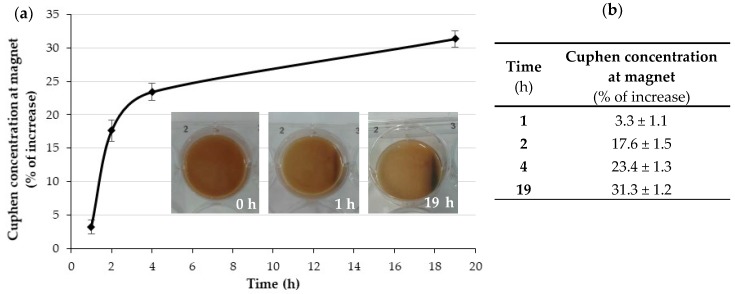
In vitro setup of the magnetic properties of Cuphen liposomes containing IONPs. (**a**) Graphical representation of Cuphen increase over time. Liposomes were exposed to a magnetic field of 560.9 mT; (**b**) Percentage of Cuphen increase at the magnet region. Results are expressed as average ± SD of at least three independent experiments.

**Table 1 nanomaterials-10-00693-t001:** Size and lipid content of liposomes co-loading IONPs and Cuphen before and after short centrifugation cycles.

DMPC:CHEMS:DSPE-PEG (57:38:5)	Ø (nm)	PDI	Lipid (µmol) (%)
**Before centrifugation**	163 ± 2	0.06 ± 0.02	10 (100%)
**Centrifugation** **15,000 *g*, 30 min**	176 ± 2	0.06 ± 0.02	5.8 ± 0.1 (59%)
**Ultracentrifugation** **42,000 *g*, 20 min**	175 ± 1	0.08 ± 0.01	5.6 ± 0.2 (54%)
Values are expressed as average ± SD of at least two independent experiments. DMPC: dimiristoyl phosphatidyl choline; CHEMS: cholesteryl hemisuccinate; DSPE-PEG: distearoyl phosphatidylethanolamine covalently linked to poly(ethylene) glycol 2000; PDI: polidispersity index. Ø: mean size of liposomes.			

**Table 2 nanomaterials-10-00693-t002:** Physicochemical parameters of Cuphen liposomes with different sizes (LIP A and LIP B) in the presence or absence of IONPs.

Nanoformulation	(Cuphen/Lip)_i_ (nmol/µmol)	(Cuphen/Lip)_f_ (nmol/µmol)	Ø (nm) (PDI)	Lipid in Pellet (%)
**LIP A**				
With IONPS	31 ± 1	22 ± 1	162 ± 1 (<0.1)	59 ± 1
Without IONPs	36 ± 2	26 ± 1	127 ± 1 (<0.1)	47 ± 2
**LIP B**				
With IONPs	38 ± 1	25 ± 1	277 ± 1 (<0.1)	80 ± 3
Without IONPs	41 ± 2	35 ± 1	236 ± 4 (<0.2)	69 ± 2
Initial lipid concentration: 30 µmol/mL. Initial Cuphen concentration: 750 nmol/mL. Centrifugation cycle: 15,000 *g*, 30 min. Values are expressed as average ± SD of at least two independent experiments; Ø: mean size of liposomes; PDI: polydispersity index.				

**Table 3 nanomaterials-10-00693-t003:** Hemolytic activity of (**a**) free IONPs and (**b**) liposomes co-loading Cuphen and IONPs. Results are expressed as average ± SD.

(a)	IONPs (mg/mL)	Hemolysis (%)	(b)	Cuphen (µM)	Hemolysis (%)
	**5.0**	3.4 ± 0.1		**200.0**	4.6 ± 1.1
	**2.5**	1.6 ± 0.3		**100.0**	3.5 ± 0.5
	**1.3**	0.7 ± 0.2		**50.0**	1.8 ± 0.1
	**0.6**	0.2 ± 0.1		**25.0**	0.9 ± 0.2
	**0.3**	0.0		**12.5**	0.4 ± 0.2
				**6.3**	0.3 ± 0.3
				**3.1**	0.2 ± 0.2

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
