# Peer review of "A Novel Hybrid Nanosystem Integrating Cytotoxic and Magnetic Properties as a Tool to Potentiate Melanoma Therapy"

_nanomaterials, 2020, doi:10.3390/nano10040693_

Round 1

Reviewer 1 Report

The paper by Reis and Gaspar and coworkers reports the preparation of lipid-based nanoformulations composed of Cuphen (cytotoxic agent) and iron oxide nanoparticles (magnetic targeting). The cytotoxicity of these nanoformulations were evaluated by MTT assay in melanoma cell lines, and the activity of each entitity, Cuphen and IONPs, were evaluated either individually as in combination. The results obtained showed that IONPs did not affect the cytotoxic properties of Cuphen. Having in mind, the use of these formulations for parenteral administration an in vivo safety hemolysis assay was performed. The results showed that these nanoformulations had negligible hemolytic effects and therefore are suitable for parenteral administration.

Regarding the characterization of these nanoformulations, the authors performed TEM and DLS analysis concerning the aggregation evaluation of magnetic nanoparticles. Considering the quantification of IONPs and Cuphen in the formulation, the authors perform indirect measurements.

This paper revealed new insights on the Cuphen formulations now associated to magnetic nanoparticles and merits to be published.

A few comments:

The text in page 8/14, after table 2, is a repetition of the text of page 7/14, please reformulated deleting the same sentences, in order to write a non-repetitive text.

The authors could evidence how magnetic targeting with IONPs will be further used in melanoma therapy. The authors presented their influence in the physicochemical/biological parameters of nanoformulations as well as their magnetic properties, but it lacks some considerations concerning real tumor treatments.

Author Response

The paper by Reis and Gaspar and coworkers reports the preparation of lipid-based nanoformulations composed of Cuphen (cytotoxic agent) and iron oxide nanoparticles (magnetic targeting). The cytotoxicity of these nanoformulations were evaluated by MTT assay in melanoma cell lines, and the activity of each entity, Cuphen and IONPs, were evaluated either individually as in combination. The results obtained showed that IONPs did not affect the cytotoxic properties of Cuphen. Having in mind, the use of these formulations for parenteral administration an in vivo safety hemolysis assay was performed. The results showed that these nanoformulations had negligible hemolytic effects and therefore are suitable for parenteral administration.

Regarding the characterization of these nanoformulations, the authors performed TEM and DLS analysis concerning the aggregation evaluation of magnetic nanoparticles. Considering the quantification of IONPs and Cuphen in the formulation, the authors perform indirect measurements.

This paper revealed new insights on the Cuphen formulations now associated to magnetic nanoparticles and merits to be published.

A few comments:

The text in page 8/14, after table 2, is a repetition of the text of page 7/14, please reformulated deleting the same sentences, in order to write a non-repetitive text.

Reply:

Thank you very much for your comment. The sentences after Table 2 were deleted (lines 272 to 289).

The authors could evidence how magnetic targeting with IONPs will be further used in melanoma therapy. The authors presented their influence in the physicochemical/biological parameters of nanoformulations as well as their magnetic properties, but it lacks some considerations concerning real tumor treatments.

Reply:

We appreciate the Reviewer’s comments. The scope of the present study was the design of a hybrid nanosystem integrating cytotoxic and magnetic properties safe for parenteral administration. The in vivo validation of our nanoformulations with magnetic properties will be performed in a xenograft murine melanoma model over 6 months up to 1 year. After establishment, we will compare the reduction on the tumor progression of induced mice receiving intravenous administrations of Cuphen liposomes or magnetic Cuphen liposomes. After each injection of selected nanoformulations, the late group of mice will be exposed to a magnetic field at tumor site. This strategy was already described in literature using docetaxel encapsulated in nanocapsules with or without magnetic properties. In a xenograft CT-26 murine model, the authors observed enhanced therapeutic efficacy using magnetic targeting compared to passive targeting (Al-Jamal et al. 2016. DOI: 10.1021/acs.nanolett.6b02261).

Reviewer 2 Report

What was the difference in size between uncoated and dextran coated IONPs? If not known, would the increase in size be in the order of a few nanometers?

A stability study on liposomes is missing. Have the authors investigated this aspect?

The authors have shown that IONPs alone do not exert a cytotoxic effect on the selected cancer cell lines? Considering that IONPs are encapsulated in the liposomes, what was the rationale for this study? Please clarify. As they have included this study, it is worth knowing how the IONPs affect endothelial cells? If they have done this study, it must be included. If not, this study is relevant. If they can justify why it is not needed, it should be supported by published literature that have studied effect of IONPs on endothelial cells (HUVECs or HAECs).

What is the size of liposomes alone i.e. without IONPs or Cuphen loading? Considering that IOPS have an average size of around 105 nm, their encapsulation in liposomes has increased the size to around 175 nm

Please cite references for the magnetism assay. Is this a standard way of doing this study? Considering that this study was done on a 6-well plate, there could be considerable variations. For example, is the position where the magnet held consistent? What prevents the solution from mixing? A schematic or a figure would be useful in understanding the experimental set-up.

Author Response

  1. What was the difference in size between uncoated and dextran coated IONPs? If not known, would the increase in size be in the order of a few nanometers?

Reply:

Uncoated and dextran coated IONPs were prepared and the morphology analysed by TEM demonstrated the high aggregation of uncoated IONPs. Based on that, we selected dextran coated NPs for all experiments in the present manuscript and, therefore, we have only analysed these nanoparticles by DLS. Nevertheless, based on TEM analysis, the increase in size of uncoated IONPs should be in the order of more than 200 nm.

  1. A stability study on liposomes is missing. Have the authors investigated this aspect?

Reply:

All the results described in this manuscript were performed with freshly prepared liposomal formulations. Nevertheless, the stability of Cuphen liposomes used in the present work has already been evaluated either in suspension or in lyophilized forms. In the lyophilized form, the stability of Cuphen in liposomes was dependent on the use of a cryoprotector (work under submission), but these studies are out of the scope of the present work. Part of these stability studies have been recently published. Using the same lipid composition included in the present work, more than 90% of Cuphen was still incorporated following an incubation period of 90 min at 37°C (Pinho et al, 2019. DOI: 10.2217/nnm-2018-0388). In addition, the stability of Cuphen in liposomes in suspension at 4°C was also evaluated and, 10 days after liposomes preparation, more than 90% of Cuphen was still associated to liposomes (Master thesis, Mariana Nave, 2013, “Nanoformulations of a potent Aquaporin-3 inhibitor with cytotoxic effect against cancer cell lines”. Faculty of Farmacy, University of Lisboa. URI: http://hdl.handle.net/10451/15294). Nevertheless, in the present work, the biological stability of the hybrid nanosytem was performed following its incubation with human red blood cells.

  1. The authors have shown that IONPs alone do not exert a cytotoxic effect on the selected cancer cell lines? Considering that IONPs are encapsulated in the liposomes, what was the rationale for this study? Please clarify. As they have included this study, it is worth knowing how the IONPs affect endothelial cells? If they have done this study, it must be included. If not, this study is relevant. If they can justify why it is not needed, it should be supported by published literature that have studied effect of IONPs on endothelial cells (HUVECs or HAECs).

Reply:

The rationale of the study was the development of a hybrid lipid nanoformulation combining cytotoxic and magnetic properties. For achieving cytotoxic properties, we have used the copper complex, Cuphen, and to achieve magnetic properties the association of IONPs was performed. As our interest is to test in vivo the so developed hybrid nanoformulation in a xenograft melanoma murine model, the in vitro assays were performed in the murine cell line B16F10, the cell line that we use to establish the melanoma model. The results obtained and presented in the manuscript revealed that IONPs did not exert any cytotoxic effect towards the murine melanoma cell line.

In relation to the interaction of IONPs with endothelial cells, there are several reports in the literature that approach this topic. For instance, in the work of Duan and colleagues (2019), dextran-coated SPIONs, tested up to the concentration of 100 μg/ml, could be internalized by human umbilical vein endothelial cells (HUVECs) without eliciting changes in cell morphology nor cytotoxic effects. Moreover, these dextran-SPIONs showed the ability to protect HUVECs from oxidative stress damages via an enhanced autophagy response (Duan et al. 2019. DOI: 10.1093/rb/rbz024). The research conducted by Matuszak and collaborators (2016) revealed that, at concentrations of 100 and 400 μg/ml, dextran-coated iron nanoparticles (IO-NP2) did not affect the growth of HUVECs, after a 72 h incubation period (Matuszak et al. 2016. DOI: 10.2217/nnm.15.216).

Although the quantification of iron content in tested formulations was not determined, for IONPs at 1 mg/ml the iron concentration should be lower than 100 µg/ml, assuming a yield of 75% for IONPs production. These concentrations are lower than the ones tested in the above published work and so toxic effects towards endothelial cells are not expected to occur.

  1. What is the size of liposomes alone i.e. without IONPs or Cuphen loading? Considering that IOPS have an average size of around 105 nm, their encapsulation in liposomes has increased the size to around 175 nm

Reply:

Taking into account the method we use for liposomes preparation for reducing and homogenizing liposomal formulations (through extrusion procedures using polycarbonate membranes with a well-defined pore size ranging from 1 µm to 50 nm), liposomes alone or loaded with Cuphen can be achieved with the same mean size. In the present work, our interest was just the comparation of the mean size of Cuphen liposomes in the absence or presence of IONPs. The obtained results are shown in Table 2. A slight increase on the mean size of Cuphen liposomes was observed for nanoformulations containing IONPs.

  1. Please cite references for the magnetism assay. Is this a standard way of doing this study? Considering that this study was done on a 6-well plate, there could be considerable variations. For example, is the position where the magnet held consistent? What prevents the solution from mixing? A schematic or a figure would be useful in understanding the experimental set-up.

Reply:

A scheme was included in the revised version of the manuscript. We have used a prototype static model where the magnet is permanent.

Reviewer 3 Report

Very important comments which I have prepared. The paper should not be published unless I can input the comments which both the editor and the authors should read. I have down loaded my comments for the authors. However, I am not able to download the same file for the editors. 

Author Response

Goal and Claims: To develop a novel hybrid nano-system for melanoma treatment, integrating therapeutic and magnetic targeting modalities. The proposed system uses a long circulating and pH sensitive liposomes loading Cuphen, a cytotoxic metallodrug, and iron oxide nanoparticles (IONPs), which at 2 mg/mL, did not interfere on cellular proliferation of murine and human melanoma cell lines.

In the Introduction, the rationale for using the current combination is logical. The toxic effects after parenteral administration of Cuphen liposomes have been considered. Thus, the current work to develop a new hybrid lipid-based nano-system by co-loading Cuphen and IONPs for magnetic targeting is reasonable. In the materials and methods section, the synthesis, preparations, characterizations and cytotoxicity studies, as well as magnetism and hemolysis assays were carried out competently. The fairly well analyzed results look interesting. Thus, publication of this manuscript can be recommended. However, this reviewer has a major concern on the fundamental approach of the current study. It is a matter of philosophy which the editor should consider.

To this reviewer, the use of toxic iron nanoparticles has to be justified, i.e., its damage to the human body must be less than its use in the present application. To start with, we note that, in J Biochem Mol Toxicol. 2018 Dec 32(12):e22225, Abakumov, et al have carried out toxicological research on IONPs by using different coated iron oxide nanoparticles. More studies should be carried out to justify the use of coated IONPs in its present application. The authors may also be interested in a recent publication on Essential Metals in Medicine: Therapeutic Use and Toxicity of Metal Ions in the Clinic, edited by Sigel (A), Freisinger, etc. in 2019. More importantly, we know that the toxicity of iron oxide nanoparticles has been known for some time. For example, in the Int J Nanomedicine, 2010; 5: 983 989, Saba Naqvi, et al, carried out concentration dependent toxicity of iron oxide nanoparticles mediated by increased oxidative stress. In the abstract, the first sentence states: “Iron oxide nanoparticles with unique magnetic properties have a high potential for use in several biomedical, bioengineering and in vivo applications, including tissue repair, magnetic resonance imaging, immunoassay, drug delivery, detoxification of biologic fluids, cell sorting, and hyperthermia.” The authors may wish to consider those results in their discussions.

The authors should address the biomechanical response of lung epithelial cells to IONPs and TiO2-NPs (Oliveira, et al. Front Physiol. 2019). There is increasing evidence that lungs can be damaged by inhalation of those NPs in environmental and occupational settings. Thus, for biomedical applications, the current manuscript should respond to those concerns. It is the opinion of this reviewer, those responses are essential for acceptance for publication in a good journal.

Reply:

We completely agree with Reviewer 3 and his/her concerns. Understanding of toxicological profiles of the nanomaterials is a crucial step for any advance in therapy. It is important to ensure that these materials are responsibly developed and safe for use, with optimization of benefits and minimization of risks.

Our manuscript is focused on one of the deadliest cancers: melanoma. The balance between benefit and risk was always considered in this work by our group and pointed out before any experiment. Like many other researchers, these concerns about the potential human and environmental health effects of nanomaterials have been expressed even by regulators and non-governmental agencies. A proof of the growing interest on this hot topic is noticeable by analysing the number of scientific papers published on toxicity of nanomaterials or related areas after 2000 . Toxicity of IONPs, although suspected to be low, cannot be properly established yet since results from in vitro studies are often inconsistent. On the other hand, the in vivo studies are limited, and human studies are almost inexistent (Valdiglesias et al. 2016. DOI: 10.1080/21691401.2019.1709855).

As a representative example, one of the first studies (year of 2000) reported the effects of the same type of IONPs that were injected in female mice. Researchers observed that the degree of morphological changes in the liver and spleen was rather low, even after a single IONPs dose that exceeds approximately by 200 times the necessary dose for diagnostics in MRI (Okon et al. 2000. Tsitologiia. 42(4):358-66).

It is evident that there is a marked difficulty to set up comparisons and establish a toxicity pattern for IONPs, mainly because of the different nanoparticles tested, different coatings, but also to the lack of methodological standardization.

The interactions of these nanomaterials with cells, as well as the potential adverse health consequences of exposure, require to be fully understood, and much work has still to be done in this field.

We have included in discussion section some references where toxicity of SPIONs was evaluated. Special emphasis to the review published by Vakili-Ghartavol and co-workers this year (Vakili-Ghartavol et al. 2020. DOI: 10.1080/21691401.2019.1709855). In this review are included some preclinical toxicity results of SPIONs and listed some commercialized SPIONs for different diagnostic and therapeutic applications.

In the present work, we assessed in vitro assays in a murine cell line, the B16F10 and preliminary toxicity using human red blood cells. The results obtained revealed that IONPs do not exert any cytotoxic effect towards the murine melanoma cell line and no hemolytic activity was observed, which might represent a very good predictor for the future studies.